# Mesoscopic superconductivity and high spin polarization coexisting at metallic point contacts on Weyl semimetal TaAs

Leena Aggarwal[1], Sirshendu Gayen[1], Shekhar Das[1], Ritesh Kumar[1], Vicky Süß[2], Claudia Felser[2], Chandra Shekhar[2] & Goutam Sheet[1]

A Weyl semimetal is a topologically non-trivial phase of matter that hosts mass-less Weyl fermions, the particles that remained elusive for more than 80 years since their theoretical discovery. The Weyl semimetals exhibit unique transport properties and remarkably high surface spin polarization. Here we show that a mesoscopic superconducting phase with critical temperature $T_c = 7$ K can be realized by forming metallic point contacts with silver (Ag) on single crystals of TaAs, while neither Ag nor TaAs are superconductors. Andreev reflection spectroscopy of such point contacts reveals a superconducting gap of 1.2 meV that coexists with a high transport spin polarization of 60% indicating a highly spin-polarized supercurrent flowing through the point contacts on TaAs. Therefore, apart from the discovery of a novel mesoscopic superconducting phase, our results also show that the point contacts on Weyl semimetals are potentially important for applications in spintronics.

[1] Department of Physical Sciences, Indian Institute of Science Education and Research Mohali, Sector 81, S. A. S. Nagar, Manauli 140306, India.
[2] Max Planck Institute for Chemical Physics of Solids, 01187 Dresden, Germany. Correspondence and requests for materials should be addressed to G.S. (email: goutam@iisermohali.ac.in).

The discovery of Weyl semimetals[1–11] facilitated the realization of Weyl fermions in condensed matter systems after more than 80 years of their theoretical discovery[12]. In quantum field theory, the Weyl fermions[13] were first shown by Hermann Weyl to emerge as solutions to the relativistic Dirac equation[3,10]. However, until the discovery of TaAs as a Weyl semimetal[3,4,6–10], such exotic particles remained elusive in nature. The Weyl semimetals are topologically non-trivial and are known to exhibit exotic quantum phenomena and unique surface states[14]. The band structure of a Weyl semimetal involves Weyl nodes, which can be imagined as a monopole or an antimonopole of the Berry curvature in the momentum space[7,15]. Each of the Weyl nodes are associated with a quantized chiral charge and the Weyl nodes are connected with each other only through the boundary of the crystals via Fermi arcs, the characterizing topological surface states in a Weyl semimetal[3,6,7,10,16]. Recently it was shown that the Fermi arcs in the Weyl semimetal TaAs are highly spin-polarized which lie in a completely 2D plane on the surface of the crystal[17]. As a consequence of such exotic topological properties, the Weyl semimetals are believed to host even richer set of physical phenomena that must be explored for better understanding of quantum mechanics and for potential device applications.

In this article, for the first time, we show the emergence of a unique superconducting phase coexisting with the highly spin-polarized surface states at mesoscopic point contacts between elemental normal metals and high quality single crystals of the Weyl semimetal TaAs. The critical temperature ($T_c$) of such superconducting point contacts was found to be 7 K.

## Results

**Point contact spectroscopy in different transport regimes.** Since a tip-induced superconducting (TISC) phase emerges only under point contacts, the traditional bulk characterization tools for characterizing superconducting phases fail to detect TISC. However, by performing point contact spectroscopy of such point contacts it is possible to explore different regimes of mesoscopic transport where the expected spectral features for superconducting point contacts are well understood. Such experiments can reveal mesoscopic superconducting phases like a TISC that emerge only under point contacts on exotic materials[18,19]. Furthermore, by driving the point contacts to the ballistic/diffusive regime of transport, it is also possible to extract spectroscopic information of the superconducting phase[20].

The experiments were performed on two crystals (crystal A and crystal B) of TaAs grown on the same batch. The crystals have been characterized by single crystal X-ray diffraction (Supplementary Fig. 3) and magneto-transport measurements (Supplementary Fig. 4). All the data presented in the main manuscript were obtained on Crystal A. Similar data was also obtained on crystal B (supplementary Fig. 7). All the point contacts presented in this work were made in the *ab*-plane of the crystals, which means the current was always injected along the *c*-axis. In Fig. 1a we show the schematic of point contacts made on the single crystals. In Fig. 1b we show a point contact spectrum between silver (Ag) and the crystal. The spectrum shows two conductance dips and a zero-bias conductance peak. Such spectra are usually seen for superconducting point contacts in the thermal regime of transport where critical current dominated effects give rise to the conductance dips[20,21]. For certain superconducting point contacts in the thermal limit, when a single physical point contact comprises of multiple electrical micro-contacts, multiple critical current

driven dips can be expected. One such spectrum is shown in Fig. 1c. The magnetic field dependence of this spectrum is shown in Fig. 1d. The spectrum smoothly evolves with increasing magnetic field and the dip structures, symmetric about $V = 0$, move closer mimicking the smooth decrease in critical current of a superconducting point contact with increasing magnetic field[20,22]. This is further illustrated in Fig. 1f where the dc current corresponding to the position of the dips (the critical current $I_c$) is plotted against magnetic field ($H$). Therefore, the point contact spectra presented above hint to the possible existence of a TISC phase on TaAs—similar to what was observed on the 3D Dirac semimetal $Cd_3As_2$ (refs 18,23) and the topological crystalline insulator $Pb_{0.6}Sn_{0.4}Te$ (ref. 19). To further confirm the existence of a TISC, we have measured the temperature dependence of the point contact resistance (Fig. 1e) corresponding to the spectrum presented in Fig. 1b. A resistive transition, similar to a superconducting transition, is clearly seen at ∼7.3 K. Moreover, the transition temperature ($T_c$) systematically goes down with increasing magnetic field, as expected for a superconducting transition. A $H_c$–$T_c$ phase diagram obtained from these data is shown in Fig. 1g. However, though the above data provide sufficient hint to a TISC phase, from the point contact data captured in a single transport regime alone does not confirm the existence of superconductivity unambiguously unless the contact can be also driven to the other regimes of transport where features related to Andreev reflection must also appear[18]. We have successfully explored these regimes as discussed below.

If the point contact are indeed superconducting, it should be possible to observe characterizing spectroscopic features in two other regimes of mesoscopic transport, namely the intermediate and the ballistic regime. In the intermediate regime, in addition to the critical current dominated conductance dips, two peaks symmetric about $V = 0$, which are known as hallmarks of Andreev reflection, must also appear. With further reducing the contact size it should be possible to transition to the ballistic/diffusive regime where the critical current driven dips should disappear and only the double peak feature associated with Andreev reflection must remain[18,20]. We have explored these two regimes and observed the expected features for super-conducting point contacts successfully. In Fig. 1h we illustrate a representative spectrum on TaAs in the intermediate regime showing signatures of both critical current and Andreev reflection. In Fig. 1i we present a representative ballistic regime spectrum where only the two-peak structure symmetric about $V = 0$ because of Andreev reflection survive and the dips because of critical current disappear[21,24]. However, the magnitude of Andreev reflection is seen to be suppressed compared with what is expected for a simple elemental superconductor. The spectrum, including the conductance suppression, could be fitted well within BTK formalism, modified to include a finite transport spin polarization[24–26]. The remarkable fit of the experimental data within the modified BTK model[27] is shown as a red line in the same panel. Therefore, from the observations made above it can be concluded that the point contacts on TaAs made with Ag tips are superconducting, though TaAs is not a superconductor. Hence the data presented above lead to the discovery of a new TISC phase on a Weyl semimetal TaAs.

**Temperature and magnetic field dependence.** To understand the nature of this new superconducting phase we concentrate on the temperature and the magnetic field dependence of the ballistic regime spectra presented in Fig. 2a and Fig. 2b respectively. The dotted lines show the experimental data points and the

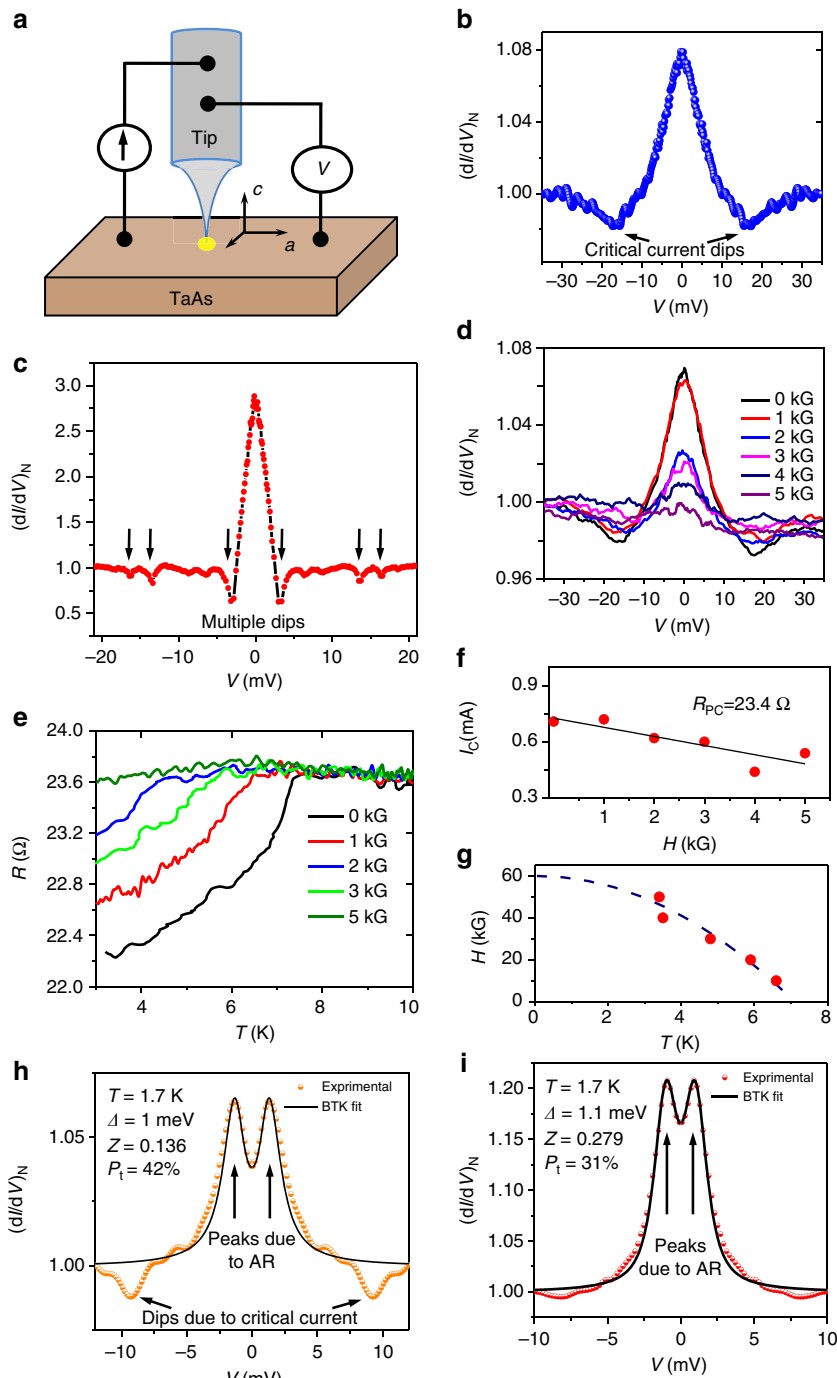

**Figure 1 | Evidence of superconductivity in three different regimes of point contact transport.** (**a**) Schematic showing a point contact between TaAs and a sharp metal tip. (**b**) A representative TaAs/Ag spectrum obtained in the thermal limit of transport at 1.8 K. (**c**) Another thermal regime spectrum showing multiple conductance dips. (**d**) Magnetic field (*H*) dependence of the spectrum presented in **b**. (**e**) *H*-dependence of point contact *R–T* showing a transition at 7 K that mimics a superconducting transition. (**f**) Magnetic field dependence of the current corresponding to the dip structure (*I*$_c$) with H extracted from the data in **d**. A linear fit to the data is also shown to highlight the decreasing trend of the critical current with increasing magnetic field. (**g**) the *H*$_c$–*T*$_c$ phase diagram extracted from **e**. The dotted line is a guide to the eye. (**h**) A spectrum obtained in the intermediate regime of transport showing the characteristic signatures of critical current and Andreev refection (AR). The orange dots are experimental data points and the solid line shows a theoretical fit using modified BTK model. (**i**) A ballistic limit experimental spectrum showing only Andreev refection (red dots) and an excellent theoretical fit to the spectrum (black line).

solid lines show the fits within the modified BTK model mentioned above. The remarkable match between the experimental data points and the theoretical fits must be noted here in both the panels. This is surprising because the superconducting phase has been derived from TaAs, which is a complex system, namely a Weyl semimetal. The gap (*Δ*) versus *T* curve extracted from the data in Fig. 2a is shown in Fig. 2c where the expected *Δ* versus *T* curve for a BCS superconductor is also shown. The maximum superconducting energy gap is found to be 1.2 meV which drops systematically with increasing

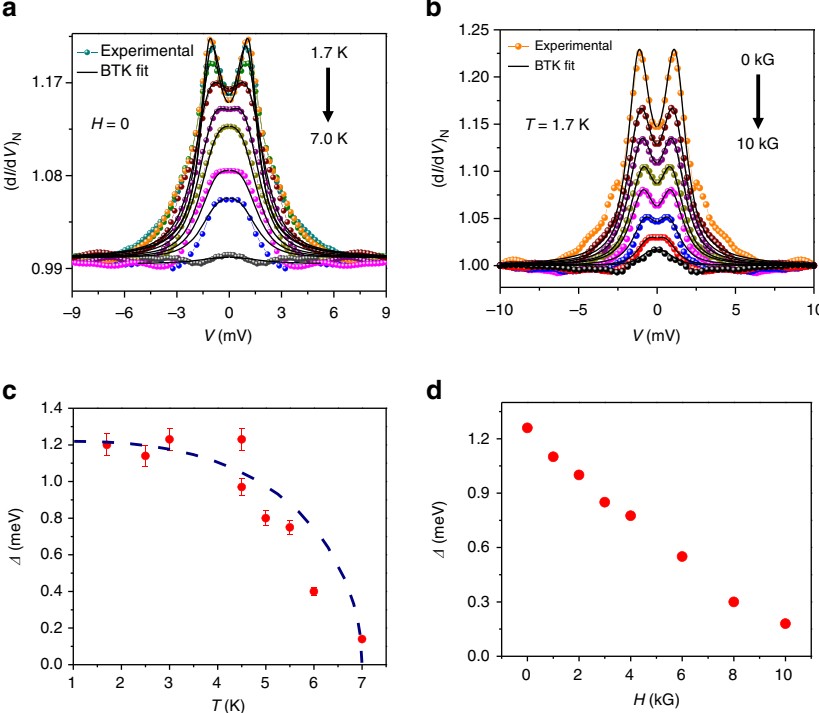

**Figure 2 | Nature of superconductivity from temperature and magnetic field dependence.** (**a**) Temperature dependence of the ballistic limit spectra (coloured dots) and their theoretical fits (solid black lines), (**b**) Magnetic field dependence of the ballistic limit spectra (coloured dots) and their theoretical fits (solid black lines) (**c**) Temperature dependence of the gap ($\Delta$). The dashed line shows the expected BCS temperature dependence. The error bars depict the range of $\Delta$ for which a reasonable fit to the experimental spectra could be obtained (see Methods section). (**d**) $H$-dependence of the gap ($\Delta$).

temperature, but does not follow the BCS line[21,22]. It is clear that the gap disappears at $T_c$ and no pseudogap-like feature above $T_c$ is observed, unlike in Cd$_3$As$_2$ (ref. 18). While the spectra could be fitted well with the conventional BTK model with spin polarization indicating the possibility of an s-wave component in the order parameter, when the deviation of the temperature dependence of $\Delta$ from the BCS and deviation of certain ballistic/diffusive regime spectra from the theoretical fits (as seen in Fig. 2; Fig. 3) are considered, the possibility of a mixed angular momentum symmetry of the order parameter where an unconventional component is mixed with a strong s-wave component emerges[28–30]. Also, the possibility of the existence of multiple gaps[31] cannot be ruled out from the set of data presented here. It should be noted that for extracting spectroscopic information we have analysed only the ballistic/diffusive limit spectra where no critical current driven dips were observed and the normal state resistance remained temperature independent.

The gap structure (double conductance peak) is seen to decrease with increasing magnetic field, as expected. Fitting Andreev reflection spectra could be non-trivial, particularly if the point contact diameter is large enough such that vortices can enter the point contact region. In the absence of an exact theory for TISC, it is difficult to estimate the coherence length. Therefore, from the presented data, it is not possible to confirm if vortices enter the point contact region and if multiple vortices can exist there. We have followed the conventional practice and used BTK theory to fit the magnetic field dependent data and found that the field dependent data could be fitted well using the spin-polarized BTK model. A plot of $\Delta$ versus $H$ as extracted from the fitting of the spectra in Fig. 2b is shown in Fig. 2d. The critical magnetic field, the field at which $\Delta$ vanishes is found to be 10 kG for the point contact presented in Fig. 2b.

**Measurement of spin polarization.** As we mentioned before, in all the ballistic/diffusive limit spectra presented above, a significant suppression of Andreev reflection was found[32]. Since for TaAs it is known that the surface states are spin-polarized[17], the suppression of Andreev reflection can be attributed to the existence of spin polarization[24,26] at the point contacts along with superconductivity. To test this effect, we have modified BTK theory to include the effect of spin polarization and the modified formalism yielded remarkably good fitting of the ballistic/diffusive limit point contact spectra with a large value of transport spin polarization up to 60%. A set of three representative ballistic/diffusive regime spectra showing high spin polarization are illustrated in Fig. 3a–c respectively. The value of the transport spin polarization is seen to decrease with increasing $Z$ as seen in Fig. 3d. A linear extrapolation of this curve leads to an intrinsic transport spin polarization of 60%. The measured spin polarization for a finite $Z$ is seen to increase with increasing magnetic field as shown in Fig. 3f. The value of spin polarization measured through this technique is lower than the 80% spin polarization measured by ARPES (ref. 17) because point contact spectroscopy is a transport measurement and in this technique the spin polarization of the transport current is measured instead of the absolute value of the spin polarization[25,27]. Nevertheless, the results and the analysis presented above indicate that the super-current flowing through the TaAs point contacts is highly spin-polarized.

**Anisotropic magnetoresistance.** To further confirm the presence of a spin-polarized current along with superconductivity at TaAs point contacts, we have performed field-angle dependence of the resistance of a ballistic point contact where the direction of the magnetic field was rotated using a 3-axis vector magnet with

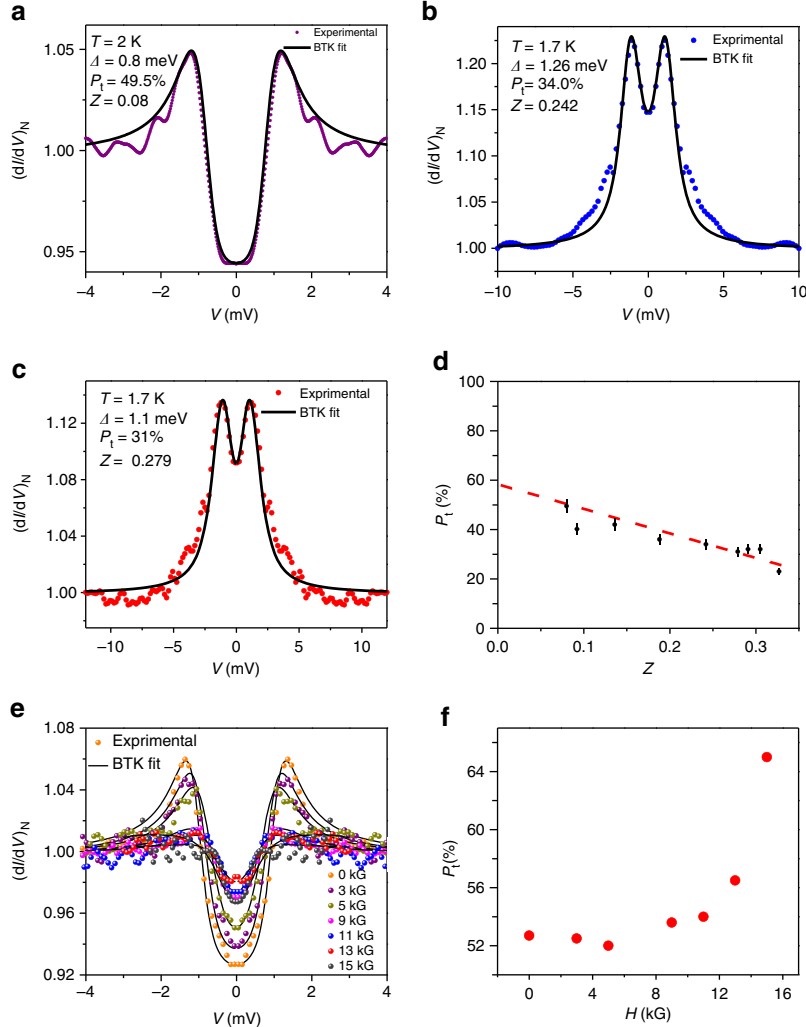

**Figure 3 | Spin-polarization measurements. (a–c)** Three representing TaAs/Ag spectra (dots) in the ballistic limit and the corresponding theoretical fits (solid lines) showing strong transport spin polarization. **(d)** Barrier ($Z$) dependence of spin polarization. A linear extrapolation of this dependence to $Z = 0$ shows a large intrinsic transport spin polariation ($\sim 60\%$). The error bars depict the range of parameters for which a reasonable fit to the experimental spectra could be obtained (see Methods section). **(e)** Magnetic field dependence of one of the ballistic point contact spectrum showing the high spin polarization. **(f)** Magnetic field dependence of spin polarization of the spectrum in **e**.

respect to the direction of current. The field-angle dependent magnetoresistance data is presented in Fig. 4a. With an applied bias $V = 13$ mV which corresponds to the normal state of the TISC, a large anisotropy in the magnetoresistance is observed which increases with increasing the strength of the magnetic field. This anisotropy in magnetoresistance can be explained if the point contact constriction is assumed to be effectively in the shape of a nano-wire and the magnetic field is rotated with respect to the direction of current flow through the nanowire. Such AMR is also seen for hybrid nanostructures involving materials having surface states with complex spin texture[33,34]. When the experiment was repeated in the superconducting state ($V = 0.3$ mV) of the TISC, as shown in Fig. 4b, the anisotropic magnetoresistance (AMR) remained equally noticeable. Therefore, the above observations conclude the co-existence of superconductivity and large spin polarization on TaAs point contacts.

To further confirm that the emergence of AMR is because of the presence of spin-polarized current, we have fitted the AMR data using the typical $\cos^2\theta$ dependence. The fitting is also shown in Fig. 4. It should also be noted that since the presented point contacts are either in the ballistic or in the diffusive regimes

of transport, where the point contact resistance is predominantly given by Sharvin's resistance which remains to be independent of the bulk resistance of the materials forming a point contact. Therefore, it is reasonable to conclude that the large AMR that we observe here is not because of the bulk TaAs.

The TISC phase presented here might emerge because of a number of reasons including local pressure, local doping and confinement effects. Since the phase could not be realized on a macroscopic interface with metallic silver films deposited on TaAs, and pressure-induced superconductivity is not known on TaAs, we believe that a combination of all the three mechanisms mentioned above could be a possible reason for the emergence of the observed TISC phase. To understand the exact mechanism behind the emergence of a TISC on TaAs, and to determine the exact symmetry of the order parameter, one possible experiment could be to use a low-temperature double-probe scanning tunnelling microscope (STM), where the first probe will be in contact with the sample to induce TISC and the second probe will perform spectroscopy in the TISC region, very close to the first tip. By moving the second tip away from the first tip while performing spectroscopy at different points, it is also possible to directly measure the

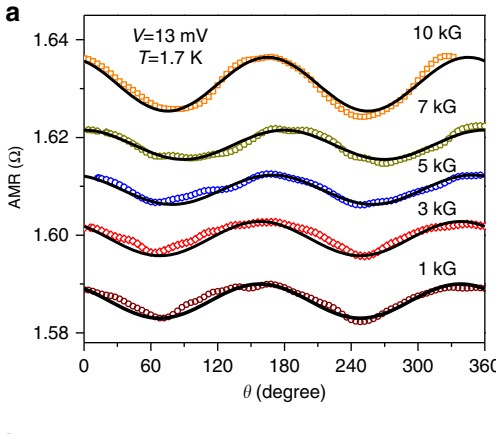

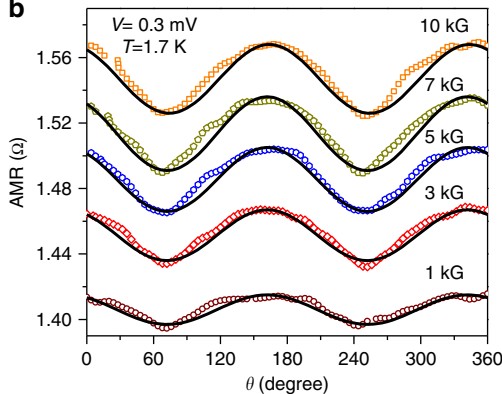

**Figure 4 | The field-angle dependence of magnetoresistance.** Anisotropic magnetoresistance of a point contact in the (**a**) normal state ($V = 13$ mV) and (**b**) in the superconducting state ($V = 0.3$ mV). The solid black lines are $\cos^2\theta$ fits.

length to which TISC extends under a point contact. Such experiments can also probe the possibility of topological and FFLO superconductivity.

## Discussion

Therefore, we have shown the emergence of a TISC phase in mesoscopic point contacts between Ag and the Weyl semimetal TaAs through transport and magneto-transport measurements at various regimes of mesoscopic transport. We used a modified BTK formalism that includes spin polarization to fit the experimental data which hinted to the coexistence of superconductivity and high transport spin polarization thereby indicating the flow of spin-polarized supercurrent through TaAs point contacts. The TISC, though shows the possibility of an unconventional pairing mechanism, no zero-bias conductance peak or a pseudogap could be observed in the ballistic/diffusive limit data. This is in contrast to the earlier observations made on a similar TISC phase on the 3D Dirac semimetal $Cd_3As_2$. To note, we noticed that another group found indication of superconductivity in TaAs point contacts[35]. Our observation of the surprising co-existence of superconductivity and high transport spin polarization at TaAs point contacts make the Weyl semimetals particularly interesting for spintronic applications.

## Methods

**Sample preparation and characterization.** All the experiments presented in this paper were carried out on high quality single crystals of TaAs. Here we provide a brief description of sample synthesis which has been presented in detail elsewhere[36]. Single crystals of TaAs were grown via chemical vapour transport. As a first step for polycrystalline material, stoichiometric quantities of Ta

(Alfa Aesar, 99.99%) and As (Chempur, 99.9999%) were weighed accurately in a quartz ampule, flushed with Ar, sealed under vacuum and heated in two consecutive temperatures of 600 °C for 24 h and 800 °C for 24 h respectively. In the next step for crystal growth, we used microcrystalline powders from step one and then added iodine (7–8 mg ml$^{-1}$) before sealing the powders in quartz tube. The crystal growth was carried out in a two-zone furnace between 900 and 1050 °C for 2 weeks. Here, $I_2$ act as a transport agent. To obtain high quality of the crystals, temperature gradient is one of the most important parameters and it varies from material to material. We optimized the temperature gradient which is 900 °C (source) to 1050 °C (sink) for TaAs. The experimental lattice parameters of the TaAs compounds is $a = 3.4310(4)$ Å, $c = 11.6252(6)$ Å, in close agreement with previous reports[37].

**X-ray diffraction.** The crystal structure of the TaAs crystal and it's orientation was determined by X-ray diffraction at room temperature. A piece of TaAs single crystal was mounted on a four-circle Rigaku AFC7 X-diffractometer set-up of Mo-Ka ($\lambda = 0.71073$ Å) radiation with a built-in Saturn 724 CCD detector. The intensity of the obtained reflections were corrected for absorption by using a multi-scan technique. The unit cell was assigned by using a 30 images standard indexing procedure. Here oscillatory images about the crystallographic axes allowed the assignment of the crystal orientation, confirmed the appropriate choice of the unit cell and showed the excellent crystal quality.

**Transport Measurements.** Temperature dependent resistivity, $\rho$ (T) of TaAs shows a metal-like behaviour in absence of a magnetic field (supplementary Fig. 4a). $\rho$ at 2 K and 300 K are 4 μΩcm and 35 μΩcm, respectively resulting in a residual resistivity ratio of 8.75. When the field is turned on, TaAs shows extremely high magnetoresistance (MR) comprising of Shubnikov de-Haas (SdH) oscillations at low temperatures (supplementary Fig. 4b). The MR values are found to be $1.5 \times 10^5$% and $1.7 \times 10^5$% at 3 and 10 K in 90 kG of magnetic field, respectively which slightly vary from 2 K to 25 K. MR is calculated from MR(%) $= 100 \times (\rho_H - \rho_0)/\rho_0$, where $\rho_H$ and $\rho_0$ are resistivity in field and 0 field, respectively. After subtracting a cubic polynomial as a background from the measured data at each temperature, the periodic oscillations in 1/H are visible up to 25 K (supplementary Fig. 4c) and their fast Fourier transform correspond to $F_\alpha = 68$kG, $F_\beta = 198$kG frequencies (see supplementary inset of the Fig. 4c). The quantum oscillation frequency is directly linked to the extremal area of the pocket (AF) by the Onsager relation, $F(= \phi_0/2\pi^2)$, where $\phi_0 = 2.068 \times 10^{-15}$Wb. The calculated corresponding area of the Fermi pockets $\alpha$ and $\beta$ are 0.00065 Å$^{-2}$ and 0.0019 Å$^{-2}$, which are consistent with the existence of tiny carrier pockets near the Fermi surface[38]. The effective mass is calculated from Lifshitz–Kosevich (LK) relation[39]:

$$\Delta \propto \exp\frac{-14.69m^* T_D}{H} 14.69m^* T/H\sinh(14.69m^* T/H)$$

where $m^*$ is effective mass, $T_D$ is the Dingle temperature, $H = 18(H = 2/(0.1 + 0.0111))$ kG is average field of taken fast Fourier transform and $m_0$ is bare mass of electron. From the above LK formula, the value of $m^*$ can be determined through the fit of the temperature dependence of the oscillation amplitude (Supplementary Fig. 4d). We find the values of $m^*_\alpha = 0.04$ and $m^*_\beta = 0.11$, which are very low and most likely ave emerged because of the linear-type of dispersion of bands. Further, the Dingle temperature $T_D$ is determined to be $\sim 6.3$ K and $\sim 4.8$ K, respectively for $\alpha$ and $\beta$ pockets and corresponding quantum relaxations time, $\tau_q (= \hbar/(2\pi k_B T_D) = 1.22 \times 10^{-12}/T_D)$ are $1.9 \times 10^{-13}$ s and $2.5 \times 10^{-13}$ s. Furthermore, measured Hall data together with resistivity show non-linear behaviour at all over temperature ranges (2–300 K) evolving the presence of two types of charge carrier which is obvious in a semimetal. After fitting Hall data by two-band model, the carrier density and mobility for electron are found to be $9.1 \times 10^{18}$ cm$^{-3}$ and $0.9 \times 10^5$ cm$^2$ V$^{-1}$ s$^{-1}$, respectively. Similarly these quantities for hole charge carrier are $8.6 \times 10^{18}$ cm$^{-3}$ and $0.7 \times 10^5$ cm$^2$ V$^{-1}$ s$^{-1}$. Extremely high MR and mobility, low fields quantum oscillations and low charge carrier density reflect high quality TaAs single crystal.

**Low-temperature measurements.** The low-temperature measurements were performed in a liquid helium cryostat working down to 1.4 K. The cryostat is equipped with a dynamic variable temperature insert (VTI) inside which there is one static VTI. The bottom part of the static VTI is made of copper for efficient cooling. The sample goes inside the static VTI which is first evacuated and then filled with dry helium exchange gas. The cryostat is also equipped with a three-axis vector magnet. The vector magnet can apply a maximum magnetic field of 6 T along the vertical direction using a superconducting solenoid and 1 T in the horizontal plane using four superconducting Helmholtz coils.

**Point-contact Spectroscopy.** Point-contact spectroscopy experiments were performed using a home-built low-temperature probe. The probe consists of a long stainless steel tube at the end of which the probe-head is mounted. The probe head is equipped with a 100 threads per inch (t.p.i.) differential screw that is rotated by a shaft running to the top of the cryostat. The screw drives a tip-holder up and down with respect to the sample. The sample-holder is a circular copper

disc of diameter 1″. A cernox thermometer was mounted on the copper disc for the measurement of the temperature. The temperature of the disc was varied by a heater mounted on the same copper disc.

The tips were fabricated by cutting the metal wire (diameter 250 μm) at an angle. The tip was mounted on the tip holder and two gold contact leads were made on the tip with silver epoxy. The samples were mounted on the sample holder and two silver-epoxy contact leads were mounted on the sample as well. These four leads were used to measure the differential resistance (d$V$/d$I$) across the point-contacts.

The point-contact spectra were captured by ac-modulation technique using a lock-in-amplifier (Model: SR830 DSP) (Supplementary Fig. 5). A voltage to current converter was fabricated to which a dc input coupled with a very small ac input was fed. The output current had a dc and a small ac component. This current passed through the point-contact. The dc output voltage across the point-contact, $V$ was measured by a digital multimeter (model: Keithley 2,000) and the ac output voltage was measured by a lock-in amplifier working at 721 Hz. The first harmonic response of the lock-in could be taken to be proportional to the differential change in the voltage d$V$/d$I$. Properly normalized d$I$/d$V$ is plotted against $V$ to generate the point-contact spectrum. The software for data acquisition was developed in house using lab-view.

**Calculation of the error bars for Δ.** Every point contact spectrum in the ballistic regime was fitted with modified BTK theory. The values of Δ and P-T were extracted as fitting parameters used in such fitting. For every spectrum, a small distribution of values of Δ and $P_T$ could be obtained for which reasonably good fitting to the experimental data was possible. First, the best fit was obtained for a combination of Δ and $P_T$, which are shown as the data points in the respective figures. After that, each parameter (either Δ or $P_T$) was varied artificially (while keeping the other one fixed) around the best fit values to find out the distribution over which reasonable fits could be obtained. The error bars in Figure 2 and Figure 3 show the small range of values around the best fit parameters for which such fitting was reasonable. The approximate fits were generated at least 10 times for estimating this range. The source of this error is mainly due to the deviation of the BTK fits from the experimental spectra at high bias that might originate from a number of possible reasons, as discussed in the main text.

**Data availability.** The datasets generated during and/or analysed during the current study are available from the corresponding author on reasonable request.

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

## Acknowledgements

The group at IISER Mohali acknowledges fruitful discussions with Professor Praveen Chaddah. R.K. thanks CSIR for junior research fellowship (JRF). G.S. would like to acknowledge partial financial support from a research grant of a Ramanujan Fellowship awarded by the Department of Science and Technology (DST), Govt. of India under grant number SR/S2/RJN-99/2011 and a research grant from DST-Nanomission under grant number SR/NM/NS-1249/2013.

## Author contributions

The point contact experiments for exploring possible TISC in TaAs were planned and designed by G.S. L.A. performed most of the experiments and took part in data analysis. S.D., S.G., R.K. and G.S. helped with the experiments and the data analysis. C.S. proposed the collaborative work on TaAs and discussed with C.F. C.S. and V.S. synthesized the high quality single crystals, characterized them and analyzed the magneto-transport data to establish the Weyl behavior of the single crystals. The analysis tools for detecting TISC were developed by S.G. and G.S. G.S. wrote the manuscript with inputs from all the co-authors.

## Additional information

**How to cite this article**: Aggarwal, L. *et al.* Mesoscopic superconductivity and high spin polarization coexisting at metallic point contacts on Weyl semimetal TaAs. *Nat. Commun.* **8**, 13974 doi: 10.1038/ncomms13974 (2017).

**Publisher's note**: 

