## [Peer Review File · Nature Communications]

Reviewers' comments:

Reviewer #1 (Remarks to the Author):

In their manuscript, the authors report on transport measurements of TaAs contacted by an Ag tip, and conclude to observe tip-induced superconductivity which seems to have a large spin polarization.

The theme of the work is certainly timely, since Weyl semimetals, and superconductivity in Weyl semimetals, are an important topic that currently interest experimentalists and theorists alike. The manuscript is well-written and clearly structured, and I find the presented data highly interesting.

I believe the authors in that they show the existence of a superconducting state in part of the sample: the resistance drops by some finite amount as a function of temperature, the temperature at which this happens lowers at increasing magnetic field, and the data is globally in reasonable agreement with the modified BTK calculation presented by the authors (in fact, I would be interested in the authors' feeling as to why this very simple approach works so extremely well, given that TaAs is a very complex material, and that tip-induced superconductivity is not clearly explained theoretically). Also, different previously known transport regimes (thermal, intermediate, ballistic) could be reached. This is indeed indicative of tip-induced superconductivity. However, there are a couple of issues and questions that the authors need to address before publication can be considered.

1) Concerning the alleged BCS s-wave nature of the gap: the authors claim the gap to be s-wave because of the temperature dependence of the gap shown in Fig. 2c. I must say that this data could be fitted with all sorts of curves, including curves that do not look at all like a standard s-wave BCS superconductor. The authors partially acknowledge this by saying that other gap symmetries or multiple gap structures might be present as well, but make a strong case for s-wave superconductivity. I feel like the data presented in Fig. 2c does not allow such a strong statement. Are there more reasons in favor of the BCS s-wave gap, especially given that TaAs as a Weyl semimetal is a topological system, and might host other types of topological superconductivity, or FFLO superconductivity?

2) How many samples were used in the study, and which plots come from which samples? How were the thermal, intermediate, and ballistic regimes reached (modification of the tip height? different samples?)?

3) Concerning the fits to the data with the modified BTK calculation: how many fit parameters were used for each plot (was it just Δ , Z , and P_t)? How consistently could those parameters be evaluated for a given sample upon varying temperature or magnetic field (e.g. in Figs 2a and b or 3e), or is there an evolution of the fit parameters with these parameters (if so, I would find it important to show that data).

4) How do the authors interpret the tunneling barrier strength Z , and its influence on the polarization?

5) Concerning the AMR data in Fig. 4: the authors give a vague explanation in terms of the possible nano-wire shape of the junction, and make an analogy to spin-textured surface states. This is a very unsatisfactory list of phenomena, and it is not clear how exactly the authors interpret their angular dependence. Is the latter due to the TaAs surface state spin structure? Is it due to the measurement geometry (and could this be tested by tilting the sample with respect to the tip)? Or do the authors not really know what is going on, and simply speculate (in which case I would encourage them to more openly state this)?

Reviewer #2 (Remarks to the Author):

Recently Weyl fermions have been discovered in TaAs crystal family, so-called Weyl semimetal. As the new topological materials, TaAs crystal family has attracted much attention in condensed matter physics and materials science. In this paper, the authors report their Ag tip induced superconductivity on the Weyl semimetal TaAs. The coexistence of mesoscopic superconductivity and high spin polarization is claimed. The observation of superconductivity is important. However, the evidence is not enough to support the finding of the high spin polarization of surface state. In the following, I have some comments which must be addressed:

1. Regarding the determination of spin polarization using point contact spectra, it is hard to distinguish the effects of inelastic scattering (interband scattering, surface defect scattering etc) and spin polarization by fitting the PCAR spectra, as detailed in Bugoslavsky et al, PRB 71, 104523 (2005), Chalsani et al, PRB 75, 094417 (2007), and Eschrig et al, PRL 111, 139703, (2013) and references therein. In this manuscript the authors didn't consider any broadening effect, which makes their conclusion less convincing. They discussed various possibilities like the mixed angular momentum symmetry of the gap parameter, and the existence of multiple gaps, but there is no solid evidence.
2. For figure 4, more detailed analyses are necessary, e.g., they didn't specify which regime it is in, and didn't explain why there is not much difference between the normal state MR with bias 13 mV and the SC state MR with bias 0.3 mV. They should also address the possibility that the AMR is just due to the bulk resistivity and not related to the point contact.
3. It is better to show some evidences such as TEM, ARPES or transport measurements (chiral anomaly etc) of the sample quality and Weyl semimetal properties.
4. Fig.1f shows strange I_c vs H curve. More explanation or description is necessary.
5. The data points in Fig.1g are not enough to give a reasonable fitting. How many fitting parameters for the curve in Fig. 1k?
6. More description and explanation are needed for the different Δ vs T and Δ vs H behaviors shown in Fig. c and d.
7. In the end of page 5, "This anisotropy in magnetoresistance ...". The authors compare their situation with nanowire situation. How about 2D?
8. Which surface is used for point contact measurement? The authors should offer the information since it is vital for the consideration of surface state.
9. The authors mention another experimental work on TaAs (ref.[35]) which appears on arXiv earlier with the dismissive description "... some possible indication of superconductivity in TaAs point contacts." Both papers report the tip-induced superconductivity on TaAs by using similar methods, which should be sure. Thus, "possible" is not appropriate, and proper description of the other work should be given.
10. In the supplementary material, data figures are needed for the part of "How did we determine the critical temperature(T_c)?" Besides, the curves in Fig.S4c and d are weird. Any explanation?
11. Some typos should be avoided. Such as "sue to" in page 4 should be "due to".

Reviewer #3 (Remarks to the Author):

The manuscript "Mesoscopic superconductivity and high spin polarization coexisting at metallic point contacts on the Weyl semimetal TaAs" by Aggarwal et al. addresses a possible tip induced superconductivity in the Weyl semimetal TaAs. The strong signatures of superconductivity coexisting with a high transport spin polarization caused by pressing the silver tip on the surface of TaAs single

crystals are found in the form of "spectral features", i.e. the point-contact conductance vs. bias voltage as a function of temperature and magnetic field. Relatively high statistics obtained on different point contacts yields consistent results. The paper brings an important message worth to be published in Nature Communications. Below I present the concerns which should be taken into account before publication.

- The authors properly distinguish between the thermal and spectroscopic regimes. The spectroscopic regimes they classify as "intermediate" and "ballistic". While the latter is standard definition instead of "intermediate" it is more common to speak as of "diffusive" (see Ref.22 of the manuscript). But the problem is not a wording. The authors discriminate the regimes by presence or absence of the critical current driven dips in dI/dV . In theory and in real experiment as well one can distinct among the three regimes even when critical current driven dips are completely absent. In the case of thermal regime the spectral features are absent, in the two spectroscopic regimes are present but with different intensity and broadening. I remark that the dips due to critical currents did not disappear completely in Fig 1(k), too.
- Of course the main issue is a low intensity of the Andreev spectra. Since the spectral features are relatively sharp, in my opinion the authors properly attribute the low intensity of Andreev reflection signal to the presence of spin polarization and applied the model (Ref. 28).
- They obtained the temperature dependence of the gap which is heavily (not slightly) scattered around the BCS curve (Fig. 2(c)). With such an accuracy to speak about unconventional components or multiple gaps is useless. Maximum one can say is that there is indication for a gap. Similarly, only very crude estimate on the critical field can be made (5-20 kG in different measurements in Fig. 1, Fig. 2, Fig. S4). The functional dependence of extracted gap vs. field is not commented at all. It is also not clear how it was obtained. What kind of model for the spectra in magnetic field was used? Point contacts can be of different size averaging or not over many vortices (are there expected here?).
- The authors modify the BTK theory to include the spin polarization (Supplement). What relation it has to the models mentioned before (Ref. 28 but also 26 and 27) and used to fit the data in Fig. 1 and 2?
- Some discussion of possible mechanism standing behind the tip induced superconductivity should be included.
- Is there any development of the spectra and the resulting parameters (gap) as a function of the point-contact resistance/size/pressure/transparency?
- The point contact is a mesoscopic object with sizes up to few microns. Why only tip induced superconductivity is observed? Could it be induced at any interface with metal, like silver or so?

REVIEWERS' COMMENTS:

Reviewer #1 (Remarks to the Author):

I thank the authors for their replies, and modifications of the manuscript. I feel like my concerns have been addressed, and am also reasonably happy with the replies to the other referees.

Since I still think that the results are very interesting and timely, I now recommend publication. As a last suggestion, the authors might replace the solid BCS line in Fig. 2c by a dotted line (for me, that would make it feel less like a fit).

Reviewer #2 (Remarks to the Author):

I am satisfied with the reply and would like to recommend the revised manuscript publish in Nature communications.

Reviewer #3 (Remarks to the Author):

The authors have considered the issues from the previous report.
With this I suggest to publish the paper in the Nature Communications.

Dear Editor,

We would like to thank you for getting our manuscript reviewed by external reviewers, for forwarding their valuable comments and for giving us the opportunity to revise and resubmit our manuscript. We are delighted to note that all the three reviewers have praised our work and have recommended publication in *Nature Communications* after addressing their concerns. We thank all the reviewers for their comments.

Below we provide point-by-point response to the comments of the reviewers:

Response to the comments of reviewer # 1

Reviewer's comment: In their manuscript, the authors report on transport measurements of TaAs contacted by an Ag tip, and conclude to observe tip-induced superconductivity which seems to have a large spin polarization. The theme of the work is certainly timely, since Weyl semimetals, and superconductivity in Weyl semimetals, are an important topic that currently interest experimentalists and theorists alike. The manuscript is well-written and clearly structured, and I find the presented data highly interesting.

Our Response: We completely agree with the reviewer that the results presented in our paper will be of great interest to the experimentalists and theorists. We thank the reviewer for highlighting the potential impact nicely. We would also like to thank the reviewer for using very kind words for our work.

Reviewer's comment: I believe the authors in that they show the existence of a superconducting state in part of the sample: the resistance drops by some finite amount as a function of temperature, the temperature at which this happens lowers at increasing magnetic field, and the data is globally in reasonable agreement with the modified BTK calculation presented by the authors (in fact, I would be interested in the authors' feeling as to why this very simple approach works so extremely well, given that TaAs is a very complex material, and that tip-induced superconductivity is not clearly explained theoretically).

Our Response: We agree with the reviewer that the results are specially surprising because a superconductor realized on a complex topological system like a Weyl semimetal could be described, to a great extent, within a simple formalism that is usually applied for conventional superconductors. At the same time, as the reviewer has pointed out in another comment, from our data it is clear that it is not completely described by the conventional theory. We also agree that a complete theoretical understanding of tip-induced superconductivity (TISC) in topological materials is lacking as of now. We believe that our present data on TaAs, where the point contact spectra could be, by and large, described by simple spin-polarized BTK theory, takes us one step ahead and reveals an important information that an s-wave component might also exist in the order parameter symmetry of TISC. We believe that as the research on TISC progresses, with more data coming up on number of topological systems showing varying symmetry of the order parameter, it will be possible to develop a consistent theory for TISC.

This, in fact, like the field of high T_c superconductivity, has the potential to reveal a new paradigm of superconductivity which is not well understood at the moment.

Following the comment of the reviewer we thought that this point should be emphasized and added one sentence in the revised manuscript: **“This is surprising because the superconducting phase has been derived from TaAs, which is a complex system, namely a Weyl semimetal.”**

Reviewer’s comment: Also, different previously known transport regimes (thermal, intermediate, ballistic) could be reached. This is indeed indicative of tip-induced superconductivity. However, there are a couple of issues and questions that the authors need to address before publication can be considered.

Our Response: We thank the reviewer again for highlighting the key spectral features acquired in different well defined regimes of mesoscopic transport which help us prove the existence of TISC beyond any doubt. We appreciate the concerns raised by the reviewer and we provide point by point response to all the concerns below:

Reviewer’s comment: 1) Concerning the alleged BCS s-wave nature of the gap: the authors claim the gap to be s-wave because of the temperature dependence of the gap shown in Fig. 2c. I must say that this data could be fitted with all sorts of curves, including curves that do not look at all like a standard s-wave BCS superconductor. The authors partially acknowledge this by saying that other gap symmetries or multiple gap structures might be present as well, but make a strong case for s-wave superconductivity. I feel like the data presented in Fig. 2c does not allow such a strong statement. Are there more reasons in favor of the BCS s-wave gap, especially given that TaAs as a Weyl semimetal is a topological system, and might host other types of topological superconductivity, or FFLO superconductivity?

Our Response: First of all, we would like to make it clear that we absolutely agree with the reviewer that based on Fig. 2c we cannot confirm pure s-wave nature of TISC in TaAs. In fact, we have not made such a claim and we have been very careful about this. We have shown the temperature dependence of the measured superconducting energy gap and the expected BCS behavior in the same panel. However, we have not concluded anything regarding the order parameter symmetry from this data. For that matter, the solid line is not a “fit” but simply an illustration of the expected BCS behavior. In order to avoid any confusion due to the way we have presented our data, we have modified the relevant text and made this point more clear in the revised version of the manuscript.

The observed superconductivity might host topological superconductivity or FFLO superconductivity. It is not possible to confirm existence/non-existence of such exotic phases from the point contact spectroscopy experiments. However, based on the reasonably good BTK fits of the ballistic regime data it is rational to conclude that at least one component of the order parameter is strongly s-wave like. To re-emphasize, this statement does not exclude the possibility of any other symmetry that might mix up with the s-wave like component.

At the end of the revised manuscript, we have also added a discussion on possible experiments that could probe the topological superconductivity or FFLO superconductivity in TISC.

Reviewer's comment: 2) How many samples were used in the study, and which plots come from which samples? How were the thermal, intermediate, and ballistic regimes reached (modification of the tip height? different samples?)?

Our Response: The studies were carried out on two single crystals. In the revised manuscript, we have named them as “crystal A” and “crystal B”. We have clearly indicated which data came from which crystal in the revised manuscript and the supplemental materials.

The point contacts were driven to different regimes by withdrawing and engaging the tip gently thereby changing the contact diameter, without changing Z considerably. The details on this are provided in the revised supplemental materials.

Reviewer's comment: 3) Concerning the fits to the data with the modified BTK calculation: how many fit parameters were used for each plot (was it just Δ , Z, and P_t)? How consistently could those parameters be evaluated for a given sample upon varying temperature or magnetic field (e.g. in Figs 2a and b or 3e), or is there an evolution of the fit parameters with these parameters (if so, I would find it important to show that data).

Our Response: As it has been described in the Supplemental materials (please see Figure S5 and the related text), the referee is right that for most of the spectra only 3 fitting parameters, namely Δ , Z and P_t have been used. For certain point contacts, a small scattering parameter Γ was also included which entered the calculation as a complex term added to the energy.

Regarding the consistency of the parameters: For the temperature and magnetic field dependent data, Z remained almost unchanged for a given point contact for the complete range of temperature and magnetic fields reported here. Δ and P_t systematically evolved with temperature and magnetic field respectively and such variations are shown in Figure 2(c), Figure 2(d) and Figure 3(e), Figure S4 (c) and Figure S4 (d) for multiple sets of data.

Following the suggestion of the referee, in the revised supplemental materials, we have shown the fitting parameters for a number of spectra (where a small Γ was added in the analysis) in a table (Table S1).

It is noted that the value of Γ remained very small for all these spectra. We would like to emphasize again, that we have obtained such spectra on more than 100 points on two different crystals but we have extracted spectroscopic information only for the spectra close to the ballistic (or, diffusive) regime (as evidenced by the spectral features and temperature dependence of the normal state resistance). The measured gap value varied from contact to

contact and the maximum measured gap was 1.26 meV.

Reviewer's comment: 4) How do the authors interpret the tunneling barrier strength Z , and its influence on the polarization?

Our Response: In the supplemental materials, we have written "Such a dependence is seen routinely in spin polarization measurements using Andreev reflection spectroscopy and is attributed to spin-flip scattering processes taking place at mesoscopic interfaces with high barrier strength." In support of this statement we have provided two references (ref. 9 and ref. 10) in the supplemental materials. In order to make the idea further clear, we have added one more reference where the theory of this is described.

Reviewer's comment: 5) Concerning the AMR data in Fig. 4: the authors give a vague explanation in terms of the possible nano-wire shape of the junction, and make an analogy to spin-textured surface states. This is a very unsatisfactory list of phenomena, and it is not clear how exactly the authors interpret their angular dependence. Is the latter due to the TaAs surface state spin structure? Is it due to the measurement geometry (and could this be tested by tilting the sample with respect to the tip)? Or do the authors not really know what is going on, and simply speculate (in which case I would encourage them to more openly state this)?

Our Response: From the observed AMR data it is clear that the point contact resistance is influenced by the direction of the magnetic field with respect to the direction of current flow. We have simply claimed that this is consistent with the idea of the existence of spin polarization at the interface. In order to show the similarities with other works where such an AMR was attributed to spin polarized current, we have provided two references and stated that a similar mechanism might also be responsible for the observed AMR here. We agree with the referee that we have not provided a rigorous theoretical explanation of the observed AMR of the point contact resistance. Neither have we claimed anything exotic like (anomalous hall effect, spin hall effect etc.) where such AMR also seen. Finding the exact origin of the AMR would be a theoretically interesting problem. However, for the main message of this paper, such an analysis is not expected to provide any additional input.

Having said that, in order to illustrate how the AMR data is consistent with the idea of the emergence of local spin polarized current at the interface, we have also shown typical $\cos^2\theta$ fits to the AMR data presented in the revised manuscript. We also show the revised Figure 4 below:

Figure 1 The field-angle dependence of magnetoresistance:}} Anisotropic magnetoresistance of a point contact in the (a) normal state ($V=13$ mV) and (b) in the superconducting state ($V=0.3$ mV). The solid black lines are $\cos^2\theta$ fits.

Response to the comments of reviewer # 2

Reviewer's comment: Recently Weyl fermions have been discovered in TaAs crystal family, so-called Weyl semimetal. As the new topological materials, TaAs crystal family has attracted much attention in condensed matter physics and materials science. In this paper, the authors report their Ag tip induced superconductivity on the Weyl semimetal TaAs. The coexistence of mesoscopic superconductivity and high spin polarization is claimed. The observation of superconductivity is important. However, the evidence is not enough to support the finding of the high spin polarization of surface state. In the following, I have some comments which must be addressed:

Our Response: We thank the reviewer for highlighting the importance of our discovery of superconductivity on the Weyl semimetal TaAs with a nice description of the context. We understand that the reviewer wants us to provide additional clarification on the observation of high spin polarization. Below we provide point by point response to the comments of reviewer # 2.

Reviewer's comment: 1. Regarding the determination of spin polarization using point contact spectra, it is hard to distinguish the effects of inelastic scattering (interband scattering, surface defect scattering etc) and spin polarization by fitting the PCAR spectra, as detailed in Bugoslavsky et al, PRB 71, 104523 (2005), Chalsani et al, PRB 75, 094417 (2007), and Eschrig et al, PRL 111, 139703, (2013) and references therein. In this manuscript the authors didn't consider any broadening effect, which makes their conclusion less convincing. They discussed various possibilities like the mixed angular momentum symmetry of the gap parameter, and the existence of multiple gaps, but there is no solid evidence.

Our Response: First of all, we agree with the reviewer that in point contact spectroscopy experiments, it is most important to perform spectroscopy in the so-called "spectroscopic regime" where, by definition, statistically, inelastic scattering processes are not allowed. Bugoslavsky et al. has nicely summarized several contact-dependent artifacts that might affect the correct estimate of the spin polarization. We would like to thank the reviewer for attracting our attention to this paper. From our experience with point contact spectroscopy, we have learned that spectroscopic information like spin polarization can be extracted only from ballistic (or diffusive) regime data where inelastic scattering processes are not allowed. Consequently, for our analysis, we have chosen the data for extracting spin polarization very carefully. We have not used any data that shows additional spectral features than Andreev reflection. We are pleased that this important aspect of our work has been appreciated by the reviewers – reviewer #3 has put special emphasis on this point saying "The authors properly distinguish between the thermal and spectroscopic regimes." We thank reviewer #3 for highlighting this point.

As we have mentioned in the revised supplemental materials, our analysis also includes an inelastic broadening parameter, Γ . However, for most of the spectra, the value of Γ remained to be either extremely small (around 1- 1% - 10% of the superconducting energy gap)

or zero. In fact, this makes the conclusions more convincing because the spectra could be fitted well with less number of free parameters (Δ , Z and P_t). In this context it must be noted that in PCAR analysis the so-called inelastic broadening parameter is included as a complex component of the energy which is expected to take care of the finite quasi-particle lifetime effect or any other inelastic processes whatsoever. All inelastic processes are dumped into a single parameter Γ without having to care for a microscopic description explaining the exact origin of Γ . As a result, for fitting PCAR spectrum, the conventional approach is to use minimum possible values of Γ , only when it is absolutely necessary. A large Γ might mean less life-time of quasiparticles thereby making the superconducting phase under the point contacts unstable (Ref: Plecenik *et al.* Phys. Rev. B **49**, 10016 (1994)).

The reason why we have chosen to include spin polarization preferentially over Γ is two-fold:

- (i) For TaAs, surface spin polarization is a known fact.
- (ii) The suppression of Andreev reflection is well explained by the introduction of a finite spin polarization. This point was also appreciated by reviewer #3. We quote from the point #2 of reviewer #3: "Of course the main issue is a low intensity of the Andreev spectra. Since the spectral features are relatively sharp, in my opinion the authors properly attribute the low intensity of Andreev reflection signal to the presence of spin polarization and applied the model (Ref. 28)." We thank reviewer #3 again for highlighting this observation in his/her report.

However, we agree with the reviewer that we have discussed about various possibilities like the mixed angular momentum symmetry and the existence of multiple gaps purely based on qualitative arguments and not providing any direct proof. Our speculations are entirely based on three factors:

- (i) small amount of deviation of the experimental data from the standard BTK fitting.
- (ii) certain small features that are not accounted for in the conventional single gap BTK theory are also seen in our spectra.
- (iii) TaAs is known to be a topologically non-trivial Weyl semimetal where a spin triplet component is expected to exist. However, the s-wave BTK model describes most part of the spectra quite well. Based on this it is reasonable to attribute the additional features to a more complex order parameter than simple s-wave. This point has been appreciated and highlighted by reviewer #1: "the data is globally in reasonable agreement with the modified BTK calculation presented by the authors (in fact, I would be interested in the authors' feeling as to why this very simple approach works so extremely well, given that TaAs is a very complex material, and that tip-induced superconductivity is not clearly explained theoretically)" and our speculations are further justified by another comment of reviewer #1: "...especially given that TaAs as a Weyl semimetal is a topological system, and might host other types of topological superconductivity, or FFLO superconductivity?"

It must also be noted that the main result of this paper is the demonstration of tip-induced superconductivity (TISC) on TaAs for which we have provided "solid evidence". Determination of the exact order parameter symmetry is certainly going to be an extremely important work that

remains to be done. However, such a work is beyond the scope of this paper and we are sure that following the publication of our paper, the community will take that up as an important problem and address the same using other spectroscopic techniques too.

Following this comment of reviewer #2, we have added a note on other possible measurements for exactly probing the order parameter symmetry. One such possible experiment to address this issue is double-probe STM at low temperature. We have started a project to build a double-probe STM where the first probe will be in contact with the sample to induce TISC and the second probe will perform spectroscopy in the TISC region, very close to the first tip. However, this will involve very sophisticated instrumentation. We have convinced our funding agency that in the light of the importance of the new TISC phase discovered on topological systems, funding such a project is justified. We have added the following text in the revised manuscript:

“To understand the mechanism behind the emergence of a TISC on TaAs, and to determine the exact symmetry of the order parameter, one possible experiment could be to use a low-temperature double-probe scanning tunneling microscope (STM), where the first probe will be in contact with the sample to induce TISC and the second probe will perform spectroscopy in the TISC region, very close to the first tip. By moving the second tip away from the first tip while performing spectroscopy at different points, it is also possible to directly measure the length to which TISC extends under a point contact.”

Reviewer’s comment: 2. For figure 4, more detailed analyses are necessary, e.g., they didn’t specify which regime it is in, and didn’t explain why there is not much difference between the normal state MR with bias 13 mV and the SC state MR with bias 0.3 mV. They should also address the possibility that the AMR is just due to the bulk resistivity and not related to the point contact.

Our Response: We have presented the data in Figure 4 as a supporting data, simply to demonstrate that the point contact resistance depends on the direction of the magnetic field and it reveals a two-fold symmetric angular dependence. This observation is very interesting and is consistent with the claim of the presence of spin polarization along with superconductivity under the point contacts. We have also shown typical $\cos^2\theta$ fit.

We thank the reviewer for his/her suggestion to provide more details about this data (like, the regime of transport etc.), which we have provided in the revised manuscript. This data was recorded for a ballistic regime point contact.

The difference in resistance that we observe in Figure 4(a) and Figure 4(b) is entirely due to the fact that for Figure 4(a), the point contact bias is above the gap and hence not superconducting and for Figure 4(b), the bias is smaller than the gap and hence the point contact is superconducting. This difference would be large if the amplitude of Andreev reflection (below the gap bias) was large. However, due to spin polarization of the point contacts, as mentioned in our paper (and reviewer #3 has highlighted this fact nicely: “Since the spectral features are

relatively sharp, in my opinion the authors properly attribute the low intensity of Andreev reflection signal to the presence of spin polarization and applied the model.”), the Andreev reflection is suppressed and consequently the difference in resistance between Figure 4(a) and Figure 4(b) is also small.

From the spectral features there is no doubt that the point contacts are either in the ballistic or in the diffusive regimes of transport, where the point contact resistance is predominantly given by Sharvin's resistance which remains to be independent of the bulk resistance of the materials forming a point contact. Therefore, it is reasonable to conclude that the large AMR that we observe here is not due to the bulk TaAs. We have added a discussion on this in the revised manuscript.

Reviewer's comment: 3. It is better to show some evidences such as TEM, ARPES or transport measurements (chiral anomaly etc) of the sample quality and Weyl semimetal properties.

Our Response: We have characterized the samples extensively before carrying out the PCAR measurements. In the supplementary materials of the first version we had mentioned that the detailed sample preparation techniques were mentioned elsewhere (arxiv: 1606.06649). The characterization data showing the signature of Weyl physics is also shown in the same preprint. In the revised supplemental materials, we have added several new data in order to address the comment of the reviewer. The new data includes magneto-transport measurements on the sample with clear observation of the resistance oscillation with magnetic field (Figure 3 of this document). We have also provided detailed analysis of the same data (which is also described below). In addition, to demonstrate the high quality of the single crystals, we have also provided single crystal X-ray diffraction (Figure 2 of this document)

XRD data: The crystal structure of the TaAs crystal and its orientation was determined by X-ray diffraction at room temperature. A piece of TaAs single crystal was mounted on a four-circle Rigaku AFC7 X-diffractometer set-up of Mo-K α ($\lambda = 0.71073 \text{ \AA}$) radiation with a built-in Saturn 724p CCD detector. The intensity of the obtained reflections was corrected for absorption by using a multi-scan technique. The unit cell was assigned by using a 30 images standard indexing procedure. Here oscillatory images about the crystallographic axes allowed the assignment of the crystal orientation, confirmed the appropriate choice of the unit cell and showed the excellent crystal quality.

Figure 2: Unit cell and x-ray diffraction of TaAs crystal. (a) a non-centrosymmetric unit cell of TaAs. (b), (c) and (d), show the rotating x-ray diffraction patterns of the TaAs crystal about the crystallographic a, b, and c-axis respectively. For each case, the rotating axis is vertical.

Transport data: Temperature dependent resistivity, ρ (T) of TaAs shows a metal-like behavior in absence of a magnetic field (Fig. 3(a)). ρ at 2 K and 300 K are $4 \mu\Omega\text{cm}$ and $35 \mu\Omega\text{cm}$, respectively resulting in a residual resistivity ratio of 8.75. When the field is turned "on", TaAs shows extremely high magnetoresistance (MR) comprising of Shubnikov de-Haas (SdH) oscillations at low temperatures (Fig. S2(b)). The MR values are found to be $1.5 \times 10^5 \%$ and $1.7 \times 10^5 \%$ at 3 K and 10 K in 90 kG of magnetic field, respectively which slightly vary from 2 K to 25 K. MR is calculated from $\text{MR} (\%) = 100 \times (\rho_H - \rho_0) / \rho_0$, where ρ_H and ρ_0 are resistivity in field and 0 field, respectively. After subtracting a cubic polynomial as a background from the measured data at each temperature, the periodic oscillations in $1/H$ are visible up to 25 K (Fig. 3(c)) and their fast Fourier transform (FFT) correspond to $F_\alpha = 68 \text{ kG}$, $F_\beta = 198 \text{ kG}$ frequencies (inset of the Fig. 3(c)). The quantum oscillation frequency is directly linked to the extremal area of the pocket (AF) by the Onsager relation, $F = (\Phi_0 / 2\pi^2) A_F$, where $\Phi_0 = 2.068 \times 10^{-15} \text{ Wb}$. The calculated corresponding area of the Fermi pockets α and β are 0.00065 \AA^2 and 0.0019 \AA^2 , which are consistent with the existence of tiny carrier pockets near the Fermi surface. The effective mass is calculated from Lifshitz-Kosevich (LK).

Figure 3. **Magneto-resistivity at various temperatures.** (a) temperature dependent resistivity, ρ (T) at 0 field and 50 kG field, (b) magnetoresistance (MR) up to 90 kG at different temperatures. A large amplitude of Shubnikov de-Haas can be seen up to 25 K, (c) (SdH) oscillations after subtracting a cubic polynomial. Fast Fourier transform (FFT) of SdH oscillations mainly gives two frequencies at $F_\alpha = 68$ kG, $F_\beta = 198$ kG (inset). (d) Effective mass corresponding to frequencies F_α and F_β .

$$\Delta\rho \propto \exp\left(-\frac{14.69m^*T_D}{H}\right) 14.69m^*T/H \sinh(14.69m^*T/H)$$

Where m^* is effective mass, T_D is Dingle temperature, $H = 18$ ($H = 2/(10+1.11)$) kG is average field of taken FFT and m_0 is bare mass of electron. From the above LK formula, the value of m^* can be determined through the fit of the temperature dependence of the oscillation amplitude as shown in Fig. S2(d). We find the values of $m_\alpha^* = 0.04$ and $m_\beta^* = 0.11$, which are very low and are likely originated to the linear-type of dispersion of bands. Further, the Dingle temperature T_D is determined to be ~ 6.3 K and ~ 4.8 K, respectively for α and β pockets and corresponding quantum relaxation time, $\tau_q (= \hbar/(2\pi k_B T_D)) = 1.22 \times 10^{-12}/T_D$ are 1.9×10^{-13} s and 2.5×10^{-13} s. Furthermore, measured Hall data together with resistivity show non-linear behavior at all over temperature ranges (2-300 K) evolving the presence of two types of charge carrier which is obvious in a semimetal. After fitting Hall conductivity by two-band model, the carrier density and mobility for electron are found to be $9.1 \times 10^{18} \text{ cm}^{-3}$ and $0.9 \times 10^5 \text{ cm}^2/\text{Vs}$, respectively. Similarly these quantities for hole charge carrier are $8.6 \times 10^{18} \text{ cm}^{-3}$ and 0.7×10^5

cm²/Vs. Extremely high MR and mobility, low fields quantum oscillations and low charge carrier density reflect high quality TaAs single crystal.

Reviewer's comment: 4. Fig.1f shows strange Ic vs H curve. More explanation or description is necessary.

Our Response: The only information that we get from Figure 1(f) is that the measured critical current shows a decreasing trend with increasing magnetic field. Following the comment of the reviewer, in order to highlight this decreasing trend, we have shown a linear fit to the presented data. The decreasing trend shows that the field dependence of the position of the resistance peaks is indeed consistent with the expected field dependence of the critical current of a superconductor. We have added a short description on this in the revised figure caption as well.

Reviewer's comment: 5. The data points in Fig.1g are not enough to give a reasonable fitting. How many fitting parameters for the curve in Fig. 1K?

Our Response: We agree with the reviewer that any attempt to fit the data presented in Figure 1(g) will not be reasonable. That is why we have not shown any fitting of the data. The dotted line shown in Figure 1(g) is just an illustration (and not a fitting) of how the H-T phase diagram for a conventional superconductor is expected from the empirical understanding. We have mentioned this in the revised figure caption. Therefore, the only conclusion that we make from this data is that the critical field decreases with increasing temperature as expected for a superconducting point contact.

Reviewer's comment: 6. More description and explanation are needed for the different delta vs T and delta vs H behaviors shown in Fig. c and d.

Our Response: Regarding Delta vs. T: As reviewer #3 has rightly pointed out, the only fact that can be concluded from the Delta vs. H curve, is that there is a gap which decreases systematically with increasing temperatures and disappears at a critical temperature. For a comparison, we have also shown a BCS temperature dependence and notice that the measured gap is scattered around the line showing BCS prediction. We agree with the observations of reviewer #1 and reviewer #3 that the data points not only deviate from the BCS line but they deviate significantly. Following these observations, we have modified the text explaining figure 2(c). We believe that reviewer #2 will agree that the revised text is more appropriate for the data.

Regarding Delta vs. H: We agree with the reviewer that we need to add more details of the Delta vs. H shown in Figure 2(d). The spectra shown in figure 2(b) show systematic magnetic field dependence, where the Andreev reflection dominated features smoothly disappear with increasing magnetic field. This indicates the smooth disappearance of the superconducting energy gap with increasing magnetic field. That is what has been highlighted in Figure 2(d) by fitting the spectra for each value of the applied magnetic field. We have added a description on this in the revised manuscript.

Reviewer's comment: 7. In the end of page 5, "This anisotropy in magnetoresistance...". The authors compare their situation with nanowire situation. How about 2D?

Our Response: From the comparison with the nanowires we wanted to qualitatively show that similar observation was also made with spin-polarized current flowing through a nanowire. Of course, such effect can be seen in a number of other systems with a spin polarized current. The idea is that the data is consistent with the claim of a large spin polarization at the interface. Following the question of the referee, we have also shown a typical $\cos^2\theta$ fit of the AMR data in the revised Figure 4 and added a discussion in the revised manuscript. Please see Figure 1 of this document.

Reviewer's comment: 8. Which surface is used for point contact measurement? The authors should offer the information since it is vital for the consideration of surface state.

Our Response: We have included this information in the revised version of the manuscript. The point contacts were made in the ab-plane of the crystal. This means, the current was injected along c-axis.

Reviewer's comment: 9. The authors mention another experimental work on TaAs (ref.[35]) which appears on arXiv earlier with the dismissive description "... some possible indication of superconductivity in TaAs point contacts." Both papers report the tip-induced superconductivity on TaAs by using similar methods, which should be sure. Thus, "possible" is not appropriate, and proper description of the other work should be given.

Our Response: We have followed the suggestion of the reviewer and removed the word "possible" in the revised version of the manuscript.

Reviewer's comment: 10. In the supplementary material, data figures are needed for the part of "How did we determine the critical temperature (T_c)?" Besides, the curves in Fig.S4c and d are weird. Any explanation?

Our Response: We have included a Figure in order to explain the T_c determination part in the revised supplemental materials. The Figure is also shown as Figure 5 of this document.

Figure 4. Scheme for T_c determination.

Reviewer’s comment: 11. Some typos should be avoided. Such as “sue to” in page 4 should be “due to”.

Our Response: We thank the reviewer for diligently reading the manuscript and suggesting to correct the typos. We have rechecked the revised version of the manuscript several times in order to fix such typos.

Response to the comments of reviewer # 3

Reviewer’s comment: The manuscript “Mesoscopic superconductivity and high spin polarization coexisting at metallic point contacts on the Weyl semimetal TaAs” by Aggarwal et al. addresses a possible tip induced superconductivity in the Weyl semimetal TaAs. The strong signatures of superconductivity coexisting with a high transport spin polarization caused by pressing the silver tip on the surface of TaAs single crystals are found in the form of “spectral features”, i.e. the point-contact conductance vs. bias voltage as a function of temperature and magnetic field. Relatively high statistics obtained on different point contacts yields consistent results. The paper brings an important message worth to be published in Nature Communications. Below I present the concerns which should be taken into account before publication.

Our Response: We specially thank reviewer #3 for appreciating the finer details of our measurements. The referee has highlighted several important aspects of our work very elegantly which helped us address some of the comments of the other reviewers. We thank the reviewer again for reading our paper very diligently and evaluating our data in the light of his/her sound knowledge of the subject, which is evident from his/her well articulated

comments.

Reviewer's comment: 1. The authors properly distinguish between the thermal and spectroscopic regimes. The spectroscopic regimes they classify as "intermediate" and "ballistic". While the latter is standard definition instead of "intermediate" it is more common to speak as of "diffusive" (see Ref.22 of the manuscript). But the problem is not a wording. The authors discriminate the regimes by presence or absence of the critical current driven dips in dI/dV . In theory and in real experiment as well one can distinct among the three regimes even when critical current driven dips are completely absent. In the case of thermal regime the spectral features are absent, in the two spectroscopic regimes are present but with different intensity and broadening. I remark that the dips due to critical currents did not disappear completely in Fig 1(k), too.

Our Response: We have followed the conventional definition of the regimes of transport. Following the comment of the reviewer we have added a discussion on this in the revised supplemental materials.

1. Ballistic regime: The contact size is smaller than the elastic mean free path. This means, the electrons statistically do not undergo scattering and therefore, the electron does not dissipate energy within the contact region. Spectroscopic studies can be possible in this regime. Both energy and momentum resolved spectroscopy are done in this regime.

2. Diffusive regime: The contact size is larger than the elastic mean free path but still smaller than the inelastic mean free path. Elastic scattering is allowed and therefore, momentum resolved spectroscopy cannot be performed in this regime. However, this is still a spectroscopic regime and energy resolved spectroscopy can be done equally well as in a ballistic regime.

3. Thermal regime: The contact size is larger than the inelastic mean free path. No spectroscopy can be performed in this regime due to dissipation.

Now, in our paper, from the spectral features, we can conclude that our point contacts were in the ballistic or diffusive regime of transport and that is why we obtained energy resolved spectroscopic information. We have also defined another regime called the intermediate regime (the term has been borrowed from Wexler's theory), where the contact size falls between the spectroscopic regime (ballistic/diffusive) and thermal regime.

Based on the above definitions, as the referee has mentioned, it is true that ballistic and diffusive regimes cannot be discriminated by the presence or absence of critical current driven conductance dips. The referee is also right that in the thermal regime, the critical current dips may not appear within the measured voltage range, if the critical current is large. However, presence of such dips in addition to the absence of the Andreev reflection dominated features helps us distinguish the spectra that are for sure obtained in the thermal regime. Now, when the dips are present along with the Andreev reflection dominated features, we cannot say that we are either in the ballistic or in the diffusive regime, as by definition, dissipation is not allowed in

these regimes. Following Wexler's formula, we called such regimes as intermediate regime. We would like to re-emphasize that the intermediate regime defined here is different from the diffusive regime. We have explained this in the revised supplemental materials.

The final remark of the reviewer on this is also appreciated. We agree that though the dip structure in our spectra has been significantly suppressed, it has not completely disappeared. However, as it has been discussed several times in the past (including in this paper: PRB 69, 134507 (2004)), the presence of such a small amount of inelastic contribution does not affect the spectroscopic results.

Reviewer's comment: 2. Of course the main issue is a low intensity of the Andreev spectra. Since the spectral features are relatively sharp, in my opinion the authors properly attribute the low intensity of Andreev reflection signal to the presence of spin polarization and applied the model (Ref. 28).

Our Response: We thank the reviewer for appreciating this important aspect of our work.

Reviewer's comment: 3. They obtained the temperature dependence of the gap which is heavily (not slightly) scattered around the BCS curve (Fig. 2(c)). With such an accuracy to speak about unconventional components or multiple gaps is useless. Maximum one can say is that there is indication for a gap.

Our Response: We completely agree with the reviewer on this point. We have modified the relevant text in order to avoid any confusion.

Similarly, only very crude estimate on the critical field can be made (5-20 kG in different measurements in Fig. 1, Fig. 2, Fig. S4). The functional dependence of extracted gap vs. field is not commented at all. It is also not clear how it was obtained. What kind of model for the spectra in magnetic field was used? Point contacts can be of different size averaging or not over many vortices (are there expected here?).

Our Response: From our data, the existence of a critical field for the point-contacts is confirmed. This is one of the most important points on the paper. Having said that, we agree with the reviewer that the estimate of the upper critical field for the point contacts presented here is an approximate value and not exact.

We have discussed the gap vs. field data in the revised version of the manuscript. The existence of vortices was not considered in the analysis as that would mean the destruction of superconductivity (at least, partially). In the existing literature, it is very common to use the BTK theory in presence of magnetic field and we have followed that. In absence of an exact theory for TISC, it is difficult to estimate the coherence length. Therefore, from the presented data, it is not possible to confirm if vortices enter the point contact region and if multiple vortices can

exist there.

Reviewer's comment: 4. The authors modify the BTK theory to include the spin polarization (Supplement). What relation it has to the models mentioned before (Ref. 28 but also 26 and 27) and used to fit the data in Fig. 1 and 2?

Our Response: Since Ref. 26 and 27 are two landmark papers on spin polarization measurement through Andreev reflection, we believe that we must cite them in this paper. We have followed the scheme proposed by Mazin in defining the term "transport spin polarization". The determination of spin polarization by first calculating the unpolarized and fully polarized current followed by an interpolation was first proposed in ref. 27. In reference 28, these ideas were applied by us to measure spin polarization of a known ferromagnet.

Reviewer's comment: 5. Some discussion of possible mechanism standing behind the tip induced superconductivity should be included.

Our Response: We believe that since TISC on topological materials is a new and extremely surprising phenomenon, providing a theory explaining the origin would take some time. Just to note, even after 30 years of the first discovery of high T_c superconductivity, no theory on the origin of high T_c superconductivity enjoys a general consensus. At this stage, every single new discovery of TISC on different topological materials is an important step forward in the direction of developing a consistent theory of TISC.

Following the suggestion of the reviewer we have included a short discussion on the possible origin of TISC in the revised version of the manuscript.

Reviewer's comment: 6. Is there any development of the spectra and the resulting parameters (gap) as a function of the point-contact resistance/size/pressure/transparency?

Our Response: We have not found any systematic dependence of the point contact spectra and the parameters (like, Δ) extracted from such spectra or the critical temperature (T_c) on the contact size (which is calculated from the point contact resistance), pressure or transparency (Z).

Reviewer's comment: 7. The point contact is a mesoscopic object with sizes up to few microns. Why only tip induced superconductivity is observed? Could it be induced at any interface with metal, like silver or so?

Our Response: The maximum point contact size that we have probed is approximately 300 nm. This has been estimated using Wexler's formula.

At present it is not understood why only tip-induced superconductivity is observed. We tried to measure after depositing thin films of elemental metals (including silver) on Cd₃As₂ and TaAs but no superconductivity was observed. From our experience, we believe three factors play important role here: (i) local carrier doping by the metallic tip, (ii) pressure applied by the tip

and (iii) confinement effects. We are designing other experiments to test each one of these separately. However, we need to build some new equipment (like, a double-probe STM) for such measurements. We have added a comment on this in the revised version of the manuscript.

Thank you.

Yours sincerely,

Goutam

Response to the reviewers

Response to the comments of Reviewer #1 :

Reviewer's comment:

I thank the authors for their replies, and modifications of the manuscript. I feel like my concerns have been addressed, and am also reasonably happy with the replies to the other referees.

Since I still think that the results are very interesting and timely, I now recommend publication. As a last suggestion, the authors might replace the solid BCS line in Fig. 2c by a dotted line (for me, that would make it feel less like a fit).

Our Response:

We would like to thank the reviewer for his/her recommendation.

We have modified Figure 2c as per the suggestion of the reviewer.

Response to the comments of Reviewer #2 :

Reviewer's comment:

I am satisfied with the reply and would like to recommend the revised manuscript publish in Nature communications.

Our Response: We would like to thank the reviewer for his/her recommendation.

Response to the comments of Reviewer #3 :

Reviewer's comment:

The authors have considered the issues from the previous report. With this I suggest to publish the paper in the Nature Communications.

Our Response: We would like to thank the reviewer for his/her recommendation.